# The Interaction between Serotonin Transporter Allelic Variation and Maternal Care Modulates Instagram Sociability in a Sample of Singaporean Users

**DOI:** 10.3390/ijerph19095348

**Published:** 2022-04-28

**Authors:** Andrea Bonassi, Ilaria Cataldo, Giulio Gabrieli, Moses Tandiono, Jia Nee Foo, Bruno Lepri, Gianluca Esposito

**Affiliations:** 1Department of Psychology and Cognitive Science, University of Trento, 38123 Trento, Italy; andrea.bonassi@unitn.it (A.B.); gianluca.esposito@unitn.it (G.E.); 2Mobile and Social Computing Lab, Bruno Kessler Foundation, 38123 Trento, Italy; lepri@fbk.eu; 3Psychology Program, School of Social Sciences, Nanyang Technological University, Singapore 636921, Singapore; giulio001@e.ntu.edu.sg; 4Lee Kong Chian School of Medicine, Nanyang Technological University, Singapore 636921, Singapore; mosestandiono@ntu.edu.sg (M.T.); jianee.foo@ntu.edu.sg (J.N.F.); 5Human Genetics, Genome Institute of Singapore, Singapore 138672, Singapore

**Keywords:** gene*environment, parental bonding, maternal care, serotonin transporter gene, rs25531, Instagram, social media, social network sites, online behavior

## Abstract

Human social interactions ensure recognition and approval from others, both in offline and online environments. This study applies a model from behavioral genetics on Instagram sociability to explore the impact of individual development on behavior on social networks. We hypothesize that sociable attitudes on Instagram resulted from an interaction between serotonin transporter gene alleles and the individual’s social relationship with caregivers. We assess the environmental and genetic components of 57 Instagram users. The self-report questionnaire *Parental Bonding Instrument* is adopted to determine the quality of parental bonding. The number of posts, followed users (“followings”), and followers are collected from Instagram as measures of online social activity. Additionally, the ratio between the number of followers and followings (“Social Desirability Index”) was calculated to estimate the asymmetry of each user’s social network. Finally, buccal mucosa cell samples were acquired, and the polymorphism rs25531 (T/T homozygotes vs. C-carriers) within the serotonin transporter gene was examined. In the preliminary analysis, we identified a gender effect on the number of followings. In addition, we specifically found a gene–environment interaction on the standardized Instagram “Social Desirability Index” in line with our predictions. Users with the genotype more sensitive to environmental influences (T/T homozygotes) showed a higher Instagram “Social Desirability Index” than nonsensitive ones (C-carriers) when they experienced positive maternal care. This result may contribute to understanding online social behavior from a gene*environment perspective.

## 1. Introduction

Among the early environmental factors, parent–infant interaction modulates human development on a biological level, influencing social, emotional, and cognitive domains [1]. Efficient parenting promotes an inclusive environment, fostering patterns for high-quality social relationships in childhood [2,3] which, in turn, can decrease behavioral and physiological distress in adulthood [4]. Simultaneously, different long-term environmental factors can play an active role in the modulation of sociality across the human lifespan. For example, children are increasingly exposed to technological devices from the first years of their lives [5] and begin to use mobile technologies before school age [6]. This exposition may contribute to social competencies learning in online and offline worlds [7]. As age limits can be easily deceived, younger and younger individuals can access social network sites (SNSs), where social exchanges between users are immediate, free, continuous, and pervasive [8]. Among these sites, Instagram (IG) is a photo- and video-sharing platform where users can follow stories, publish or comment on posts, watch or upload content, and chat privately. Some evidence has demonstrated that increased Instagram activity might be associated with positive outcomes, acting as a motivator for a satisfying self-presentation and increasing self-confidence [9]. However, it is also associated with negative consequences, such as a risk of developing of a depressed mood [10] and a detrimental effect on mental well-being [11].

The influence of environmental cues might not fully explain such a tangled phenomenon as social behavior. According to the biopsychosocial model of wellness and illness, it is more likely that genetic predispositions could be crucial in shaping the effect of early experiences on social conduct on SNSs. At the molecular level, individual behaviors result from genetic predispositions and environmental stressors or conditions, especially during critical periods [12]. According to the *differential susceptibility model* [13], a given genetic predisposition is not uniquely “good” or “bad” in human behavior. Indeed, the level of susceptibility to life events is flexible and regulated by the allelic expression involved in the physiological and behavioral responses to a range of triggering environmental hazards. In the myriad of biological regulators, serotonin is a neurotransmitter implicated in the neural circuits of emotional regulation, social functioning, and social affiliation [14,15]. Serotonin levels may vary across brain regions and affect individuals differently [16]. The serotonin transporter encoded by the serotonin transporter gene (SLC6A4) is the primary regulator of serotonin removal from the synaptic clefts. Specifically, the variable number of tandem repeats (VNTR), known as the promoter region of the serotonin transporter gene-linked polymorphic region (5-HTTLPR), and the A/G single-nucleotide polymorphism (SNP) located within the 5-HTTLPR repeats (rs25531) [17,18] have been investigated in association with perceived attachment and parental attitudes [19,20,21,22]. Concerning 5-HTTLPR, for instance, evidence from studies on animal models suggests that variations in the polymorphic region of this gene confer greater susceptibility to environmental factors. For instance, rhesus macaques carrying the short-allele variation benefited from a supportive and caring environment during early interactions, showing greater social competence later in life [23]. As for human models, Truzzi et al. [24] discovered that men with a higher predisposition (carriers of the short form of 5-HTTLPR) for sensitivity to maternal overprotection showed a distressed heart rate to female cry. Concerning rs25531, two allelic forms have been found: the substitution of thymine (T) and cytosine (C), with the paired nucleotides guanine (G) and adenine (A), shows differential susceptibility to stressors [25,26,27,28]. However, until now, literature has not been able to disentangle which variation is more associated with adaptive social responses [29,30]. Genetic vulnerability conferred by the A/A and T/T genotype, whose transcription processes are associated with a lower reuptake of serotonin, can be moderated by the social support of parents [31]. Social support in childhood influences the risk of mental disorders in adulthood and facilitates a wide range of possible behavioral outcomes [26,28,32,33,34]. For instance, people with a history of low-quality parental support in childhood could show maladaptive behaviors and benefit from others’ support in adulthood [35,36,37,38]. Conversely, individuals with the C or G allele showed enhanced serotonin degrees combined with lessened sensitivity to pain, traumatic events, and care [27]. Within the social context, higher social avoidance and lower emotional engagement were related to a decrease in serotonin, whereas higher prosocial behavior and lower social anxiety from childhood were related to an increase in serotonin [20,26]. A piece of experimental research highlighted a positive association between avoidance towards peers and neural activity of the anterior prefrontal cortex in response to distress stimuli for C-carriers but not for T/T homozygotes [39].

In summary, we could assert that sociability emerges from the combined effects of nature, driven by genetic and biological predispositions and nurture (e.g., social environment, social education across development, and social experiences with parents and peers) [12]. However, the function of the “sociability genes” (i.e., rs25531) according to the early environment’s quality on the online social interaction of adults has not yet been investigated.

This research explores how the interaction between the serotonin transporter gene and the individual’s relationship with caregivers during childhood modulates adults’ online social interactions on Instagram. Our attention is focused on Instagram, one of the most popular social networks, to unveil online social interaction’s fundamental mechanisms. Instagram users were firstly required to fill the *Parental Bonding Instrument* to measure their recalled parental bonding. Buccal mucosa samples were subsequently collected, and one 5-HTT single-nucleotide polymorphism (SNP: rs25531) was examined as a genetic factor. Finally, three social media indexes were assessed from the Instagram profile of each user: the number of (i) published posts as an index of the social productivity of the network, (ii) number of followed users (here called “followings”) as a measure of prosocial activity, (iii) number of followers as a measure of the social attraction of other users. A fourth Instagram index was obtained as a proxy of asymmetry of the social network, calculated through the ratio between the number of followers and the number of followings: (iv) the “Social Desirability Index” (SDI).

For each Instagram index as the dependent variable, we hypothesized an interaction effect between the genetic component and the attachment scores, independent of the gender. More precisely, we expected that adult Instagram users with a genetic risk factor (rs25531 T/T) who experienced a supportive and positive past relationship with parents (High Parental Care, Low Parental Overprotection) would show a greater online social activity compared to protective genetic carriers (rs25531 C-carriers). Although we formulated a directional hypothesis, we could not predict *a priori* on which Instagram variable potential effects would have been displayed. Given the new role of Instagram activity from a gene–environment perspective, we adopted an exploratory approach.

## 2. Methods

### 2.1. Participants

Sixty-one (*N* = 61) young adults participated among the students at Nanyang Technological University (Singapore). Inclusion criteria were as follows: (i) no history of genetic, neurological, or psychiatric disorders; (ii) age lower than 30 years old; (iii) owning and using an Instagram account. Participants with incomplete Instagram data or partial online compilation of the questionnaire were omitted (*N* = 4). Thus, our final sample included 57 Singaporean nonparent adults (16 males and 41 females) aged 18–25 years old (*M* = 20.82, *SD* = 1.59) (Table 1).

### 2.2. Procedure

Participants were recruited through the undergraduate research participation pool from Nanyang Technological University of Singapore (NTU). The exclusion criteria adopted were having children (since parenthood could alter the recalled parental bonding in childhood), not owning an Instagram account, or having a history of mental health issues. Students could voluntarily book their participation in the study using the School of Social Sciences Research Participation System System of the University. After completing the whole procedure, participation in the study was rewarded with 1 credit as a contribution to a psychology module’s course assessment, according to the program of NTU’s School of Social Science. The exclusion criteria adopted were having children since parenthood could alter the recalled parental bonding in childhood, not owning an Instagram account, or having a history of mental health issues. The study was approved by NTU’s Institutional Review Board. The methodology was performed under the relevant guidelines and regulations. Before collecting data, participants provided informed consent and reported demographic details. The study included three assessments: (1) participants completed the self-report questionnaire *Parental Bonding Instrument* on the web-based survey platform Qualtrics and (2) supplied the link of their Instagram page from which a Python program extracted four parameters of their online activity. When the Python program failed, Instagram data were assessed manually. After the information from the questionnaire and Instagram account were completed, (3) a sample of buccal mucosa was collected from each participant and genotyped in the laboratory.

### 2.3. Parental Bonding

The *Parental Bonding Instrument* (PBI) [40] is a 50-item self-report questionnaire administered to adults to quantify parental bonding during childhood and adolescence. This instrument allows participants to recollect the quality of the relationship with their parents, rating their caregiving behavior on a 0–3 Likert scale. The PBI (averaged Cronbach’s α = 0.88 in our sample) estimates the reported early attachment across four subscales: maternal and parental care, maternal and parental overprotection. The two dimensions of care evaluate the level of parental closeness and affection (e.g., “*Could make me feel better when I was upset*”). Conversely, the two subscales of overprotection measure the level of parental imposition or detachment (e.g., “*Made me feel I wasn’t wanted*”). The PBI is widely used worldwide with reliable psychometric properties in terms of high validity, internal consistency [41,42], and its use in previous gene*environment research paradigms [24,43,44].

### 2.4. Instagram Parameters

We automatically extracted Instagram data from the participants’ profiles by adopting an *ad hoc* script created with Python based on ‘*beautifulsoup*’ library. We selected four variables from Instagram data: *number of posts*, *number of followings*, *number of followers*, and *Social Desirability Index*. A large number of studies have focused on the frequency of online behavior on social media platforms by collecting information about the number of posts, followings, and followers or friends [45,46,47,48,49,50]. These indexes have been a basic and stable feature of the Instagram platform across the different versions and independently of the smartphone’s operating system. Specifically, the number of posts describes the quantity of published and shared material on the participant’s profile. On Instagram, posts can usually be pictures, videos, or graphical texts—all content that other users can actively comment on and like [51]. This parameter suggests the users’ level of productivity and desire for exposure to others’ judgments. The number of posts was also associated with the user’s psychological vulnerability (i.e., depression) [52]. The number of followings represents the number of followed users by an assessed study participant. Instagram users compare their behavior with other users who become the reference points of their virtual activity. More followings demand more social interactions in terms of greater time invested in online social activities and higher passive exposure to content published by other users [53,54]. The number of followers is the number of users who follow the assessed study participant. Followers’ recognition and support act as a reward for the user’s online behavior, who then adjusts his/her social activity in line with the social network’s requests to boost agreeableness (i.e., the number of likes) [55]. The Social Desirability Index is obtained by the ratio between the number of followers and followings. We calculated this ratio to explore the asymmetry between these two parameters in each participant’s network. As discussed in a previous work [44], this parameter can reveal “*the tendency of some Instagram users to maximize the number of followers at the expense of the number of followings*”.

No sensitive data or media content were extracted or collected during the scraping process to ensure the privacy of the participants.

### 2.5. Genotyping

This study adopted the same DNA sequencing technique described by Bonassi and colleagues [43]. DNA extraction and genotyping were performed by ACGT, Inc. (Wheeling, IL, USA) for the rs25531 region target (gene SLC6A4). DNA was derived from each kit, applying the Oragene DNA purifying reagent. DNA concentrations were evaluated through spectroscopy (NanoDrop Technologies, Wilmington, DE, USA). A polymerase chain reaction (PCR) reaction of 20 ll including 1.5 ll of genomic DNA from the test sample, PCR buffer, 1 mM of each primer (forward 5${}^{\prime }$-GGCGTTGCCGCTCTGAATGC-3${}^{\prime }$ and reverse 5${}^{\prime }$-GAGGGACTGAGCTGGACAACCAC-3${}^{\prime }$), 10 mM deoxyribonucleotides, KapaTaq polymerase, and 50 mM MgCl_2_ was administered. PCR operation consisted of, first, a 15-min denaturation at 95 ${}^{{}^{\circ}}$C and 35 cycles at 94 ${}^{{}^{\circ}}$C (30 s), 60 ${}^{{}^{\circ}}$C (60 s), and 72 ${}^{{}^{\circ}}$C (60 s), then a further 10-min lengthening at 72 ${}^{{}^{\circ}}$C. PCR reactions were genotyped with an ABI 3730xl Genetic Analyzer (Applied Biosystems Inc., Waltham, MA, USA) and standardized with GeneScan 600 LIZ (Applied Biosystems, Inc.) size standards on each sample. Genotypic data were analyzed by GeneMapper ID (Applied Biosystems, Inc.).

The average distribution of the genotypes in the Asiatic population is 86–87% for T/T homozygotes and 13–14% for C-carriers (1000 Genomes project, BioSample: SAMN07486024, dbSNP (Short Genetic Variations), 2017), whereas the distribution in our sample was 73.68% for T/T homozygous (*N* = 42), 26.32% for C/T heterozygous (*N* = 15) and 0% for C/C homozygous (*N* = 0). This genotype distribution follows the Hardy–Weinberg Equilibrium ($X^{2}$(1) = 1.31, ns). Participants having one C allele (C/T) were classified into a single C-carriers group. Participant gender ($X^{2}$(1) = 0.23, ns) did not significantly differ between the two groups, T/T vs. C.

### 2.6. Analysis

Data were analyzed using R (version 4.0.0). Instagram variables were first standardized with z-scores (Table 2). Next, measured variables’ distributions were checked for normality, skewness, kurtosis, presence of outliers and were then visualized by density and quantile–quantile plots (Table 2).

The distance of each observation to the centroid was estimated to detect outliers defined as a value equal to 2 *SDs* above/below the mean [56]. Three extreme values were identified as outliers for the number of followings and the number of posts, two values for the SDI, and one value for the number of followers out of 57 observations for each Instagram variable. They were replaced with the mean value in observations, excluding the outliers. Each Instagram variable showed a normal univariate distribution. Only the number of posts did not present a Gaussian distribution, and a logarithmic transformation was applied to allow for the adoption of parametric tests. Moreover, assumptions of the homogeneity of variance and multicollinearity across the sample were verified.

A preliminary analysis was executed on Instagram variables to exclude any effect of participants’ gender (4 repeated measures, α = 0.0125). An equivalent statistical procedure was adopted for Instagram variables overall for our hypothesis-driven analysis. Four multiple linear regressions were performed with the Instagram value as the dependent variable, the 5-HTT gene genotype rs25531 (T/T and C carriers) as a between-subjects factor, and the PBI dimensions (i.e., maternal care, maternal overprotection, paternal care, and paternal overprotection) as continuous covariates. Hence, a Bonferroni correction was applied for the number of tests (4 repeated measures, α = 0.0125). Post hoc power analysis was calculated with *G*Power* software (version 3.1) [57]. One 2-way interaction effect was obtained and represented by a bar plot and a scatterplot with linear models. Pearson’s *r* and Fisher’s *z* coefficients were used to estimate the covariates’ effect on the dependent variable, while $R^{2}$ and Cohen’s *d* were calculated to quantify the size of the significant effects.

For the significant interaction between the genotype and the PBI covariate on the Instagram variable, post hoc Student’s *t*-tests (2 repeated measures, α = 0.025) were performed within the low vs. high PBI groups (obtained by the median split procedure) to prove differences between the two genetic clusters.

## 3. Results

Means and standard deviations of the measured variables are reported in Table 3.

### 3.1. Instagram Variables: Preliminary Results

Four two-tailed Student’s *t*-tests were conducted (see Table 4) to exclude any influence of gender on Instagram variables. Only the number of followings was higher in males than females (*t* = 2.60, df = 55, *p* < 0.0125). Consequently, participants’ gender was included as a between-subjects variable limited to the number of Instagram followings.

### 3.2. Instagram Interaction Effects

#### 3.2.1. Social Desirability Index

Results of the regression analysis on the standardized SDI (*F*(9,47) = 1.92, $R^{2}$ = 0.27, power = 0.83) revealed that a significant interaction between maternal care and genotype emerged for Instagram SDI (β=0.04, SE = 0.01, *t* = 2.86, *p* < 0.006). The distribution of genotypes TT vs. C-carriers was not significantly different between high vs. low maternal care ($X^{2}$(1) = 2.46, ns). No main effect of covariate, genotype, or other interactions with genotype were significant. Maternal care was negatively associated with the Instagram SDI for C-carriers (*t*(15) = −2.50, df = 13, *r* = −0.57, *p* < 0.03), but positively associated with the Instagram SDI for T/T homozygous (*t*(42) = 2.75, df = 40, *r* = 0.40, *p* < 0.01) (Figure 1). A significant difference between the slopes for T/T and C-carriers was confirmed by Fisher’s *z* (*z* = 3.24, *p* < 0.001). Moreover, one-tailed post hoc Student’s *t* tests on the C-carriers vs. T/T in low and high maternal care groups revealed that the SDI was significantly different between C-carriers and T/T homozygotes only when they reported past experiences of high maternal care (*t* = 2.30, df = 25, *p* < 0.025), but not when they had a past of low maternal care (*t* = −0.76, df = 28, ns) (Figure 1). Homogeneity of variance of the SDI by maternal care was checked ($K^{2}$ = 0.12, df = 1, ns).

#### 3.2.2. Number of Instagram Posts, Followings, and Followers

No significant main effect or interactions were found.

## 4. Discussion

In this study, we probed how alleles in rs25531 interact with recalled parental care and overprotection in the modulation of social behavior on Instagram. Specifically, we hypothesized one genotype*environment interaction (rs25531 SNP*parental bonding in childhood) on Instagram variables.

In accordance with the hypothesis, we discovered that adult Instagram users with genetic vulnerability (T/T homozygotes) showed varying Instagram social conduct related to the quality of the parental practices experienced in childhood. As expected, T/T homozygotes who reported high maternal care scores exhibited an increasing trend in the Instagram SDI. In contrast, those who reported low scores in the equivalent dimension determined a decreasing trend in the same index. Interestingly, C-carriers showed the opposite pattern: as maternal care scores were lower, Instagram SDI increased. In particular, vulnerable genetic carriers (T/T homozygotes) with a positive relationship with the mother displayed a higher Instagram SDI than protective genetic carriers (C-carriers).

This study investigated the *differential susceptibility model* on rs25531 within a novel and innovative framework: the online social behavior on Instagram—a SNS. In line with the model [13], a T versus C allele’s presence implicated a conditional variation in the online social response: T/T homozygotes who benefited from efficacious maternal care displayed a higher Instagram SDI than C-carriers. In this condition, Instagram users carrying the C allele were less susceptible than T/T homozygotes to supportive maternal care during childhood. Compared with T/T genotype, the C allele conferred lower vulnerability to adverse conditions linked to neglectful caregiving and less advantage from a favorable relationship with the mother. Consequently, users with the C allele exhibited an Instagram social response unconcerned about the advantages promoted by appropriate maternal care [27,58]. Otherwise, users with the T/T genotype took advantage of a protective and comfortable maternal relationship, exhibiting higher asymmetry between the number of followers and followings than less-sensitive ones [59]. As a result of developmental flexibility, the T allele predisposes individuals to be more vulnerable or plastic according to the functional or dysfunctional attachment with the mother [60,61,62].

The present research extends previous evidence of genetic influence on the frequency of online social behavior [63,64,65]. Regarding rs25531, a previous study found that carriers of susceptibility genetic factors (T/T homozygotes) presented a higher Instagram number of followings than nonsensitive ones (C-carriers) when they reported a high level of confidence toward others [66]. As pointed out by another work [44], in the current study, users genetically susceptible to environmental influences displayed higher SDI when they had previously experienced positive maternal bonding. However, recent work on rs25531 disclosed an association between avoidance in adult attachment and Instagram number of followings independent of genetic predispositions [67].

### 4.1. Possible Implications in Brain Mechanisms

The increased vulnerability conferred by rs25531 could influence the regulation of 5-HTTLPR expression and, in turn, processes mediated by serotonin and serotonergic pathways, such as sensory processing of socially relevant stimuli and emotional decision making [68]. Serotonin neurons are particularly dense in the prefrontal cortex (PFC), suggesting that it acts as a major modulator of the functions carried out by this area [69]. The PFC is closely linked to the amygdala and is involved in processing information related to social perception and experience. A targeted region of the PFC, namely, the orbitofrontal cortex (OFC), is involved in operations requiring social flexibility, such as evaluating rewards and risks when responding to a social situation and processes of salient social cues [70,71]. In fact, OFC integrates information regarding environmental context, emotions, and memory [72]. OFC is also shown to be involved in mental activity aimed at suppressing negative thoughts, which is strictly linked to the perception of parental care received at the early stages of life [73]. Other than the PFC, serotonin is involved in the circuits of the nucleus accumbens (NAc) related to social rewards [74] and neural mechanisms of the medial preoptic area (MPOA) of the hypothalamus [75] that support maternal behavior [76]. As such, serotonin-related regulations appear to be widely involved in planning emotional and behavioral responses to environmental social stimuli [39,72,77], especially when the social incentives are mediated by the expectations of the self and of others [28].

### 4.2. The Importance of Social Desirability

We focused our attention on the SDI because it represents the distribution of contacts for each user on Instagram. This index attributes a size to each user’s network structure, where the number of followers increases at the cost of the number of followings. Maximizing the SDI means enlarging the network structure to achieve greater approval and benefit from the masses [78,79]. Within this frame, we argue that higher maternal care could determine the basis of new prosocial behavior in the offline environment, later combined with a greater need for social desirability in the online environment [9,44,80]. An Instagram user sensitive to early maternal experiences could, in turn, respond more efficiently to the request of others and support social exchanges in response to their followers’ expectations [55,81,82]. This outcome suggests that the perception of maternal warmth could favor or reduce needs and abilities related to self-management [83], influencing the capacity to correspond to one’s followers’ expectations [84]. SDI resents the user’s online activity in terms of frequency, minutes, or hours of usage per day, and time passed since the subscription to the platform; these parameters have not been investigated in the present study, but future research could elucidate the contribution of time-related variables to the SDI. In fact, SDI presents the potential for a solid link to the posting or usage activity since they might reinforce one another. Additionally, if the user is not competent or familiar with the usage of Instagram, the profile indexes might display lower numbers of following or followers. In comparison, more expert users could interact more efficiently, reaching more significant numbers of following and attracting more users’ requests for connection. A limit of the SDI is that numbers of following and followers can be easily manipulated by the profile owner, for instance, by deleting following accounts to maximize the impact of the number of followers.

## 5. Related Works

This study represents a further step of a series of investigations involving Instagram usage within a gene-by-environment frame. Environmental factors included recalled parental bonding, psychological mechanisms underlying interpersonal behavior, and experience in adult intimate relationships, while genetic information concerned targeted polymorphisms of the serotonin transporter gene (rs25531 T/C) and the oxytocin receptor gene (rs53576 G/A; rs2254298 G/A). Table 5 presents comparative results across the studies related to the present work. The sample analyzed for the studies included in the table is analogous to the present work (see Table 1) and followed the same experimental design. Looking at the results emerging for the studies included in the table, our findings were consistent with previous results on rs25531, which suggested a large variability between carriers in the behavioral response to social and distressing stimuli [85,86]. Specifically, rs25531 appears to modulate the sensitivity to environmental factors related to social expectations towards significant others more than levels of anxiety or avoidance in close relationships. Moreover, results display the moderating role of the genotype on the sensitivity of the phenotype to environmental influences [26] in support of the genetic susceptibility to the beneficial versus dysfunctional effect generated, respectively, by enriching or undermining circumstances [12,13].
ijerph-19-05348-t005_Table 5Table 5Comparative table of works related to Instagram usage from a gene-by-environment perspective. *OXTr* = oxytocin receptor gene.*N*ReferenceGenetic FeatureEnvironmental FactorResults1Bonassi et al., 2020a [44]*OXTr* rs2254298 (A/G)Maternal Care; Maternal Overprotection; Paternal Care; Paternal OverprotectionA-carriers for *OXTr* rs2254298 with low paternal care showed a fewer number of posts on Instagram; (ii) effects of the interaction between *OXTr* rs2254298 and maternal overprotection on SDI scores2Bonassi et al., 2020b [66]rs25531 (T/C)Confidence; Need for approval; Relationships as secondary; Worry about relationships; Discomfort with closenessGreater confidence levels in rs25531 T/T carriers are associated with greater number of followings.3Bonassi et al., 2021a [67]rs25531 (T/C)Avoidance; AnxietyAbsence of gene-by-environment interaction; association between Avoidance and “followings”4Carollo et al., 2021b [87]*OXTr* rs53576 (A/G)Avoidance; AnxietyA/A carriers for *OXTr* rs53576 showed greater numbers of “followings” than G-carriers independent of the anxiety or avoidance levels5Present studyrs25531 (T/C)Maternal Care; Maternal Overprotection; Paternal Care; Paternal OverprotectionEffects of the interaction between T/T and positive maternal care associated with greater values of SDI

## 6. Conclusions

This research is not exempt from some limitations. Firstly, the low sample size was constrained by data collection of multiple variables in a fixed period. Data collection was circumscribed to young adults between 18 and 25 years old from the same cultural context. Even though the 1000 genomes project found no significant differences in the allele variability of rs25531 (T > C) between Asian populations (1000 Genomes project, BioSample: SAMN07486024, dbSNP (Short Genetic Variations), 2017), an unbalanced sample composition should be noted (e.g., *N* = 56 Chinese and 1 Indian). At a cultural level, Singapore is a multiethnic country, where citizens of different ethnic backgrounds (e.g., Chinese, Indian, Malay) share a sense of belonging to the community [88]. Moreover, an unbalanced rate in the ethnicity of Singaporean participants may reflect the demographic distribution of the country since Chinese citizens represent over 75% of Singapore’s population [89].

Overall, future studies could overcome these potential limitations. For instance, collecting data from a cross-cultural sample (e.g., Western- vs. Eastern-oriented) could allow for inspecting differences in the genetic predispositions, caregiving propensities, and Instagram behaviors. Instagram responses could also be explored to prove that online communication frequency and duration could vary between age ranges (e.g., adolescence, early adulthood, adulthood, old age).

The participants’ socioeconomic status (SES) was also not evaluated. Users who live in a situation of economic security could increase their Instagram activity to prove their ideal prosperity to their followers [90]. The year of first registration on Instagram was also not collected; thus, the indexes’ quantities were not controlled for the duration of online activity. Furthermore, the analysis was performed on no more than three Instagram parameters and one combined index that did not discriminate “bot accounts” or “artificial” followers that can be purchased instead of being earned through social capital [91,92]. Future investigations should discriminate and analyze data only from real followed members and no other reel, advertisement, paid promotions, or other content suggested automatically by the social media algorithm. Finally, the recalled parental bonding was measured with a self-reported instrument.

Future studies should consider a more extensive and multicultural sample to probe potential cross-cultural differences in Instagram behavior and a gender-balanced sample including participants of different ages. The assessment of plausible moderators of Instagram usage such as the SES and the sense of membership with peer groups may also be considered. The increase in the number of posts, followers, and followings should be traced in relation to the year of the first registration on Instagram. Novel Instagram variables and indexes able to detect the variety of the shared content may also be taken into account, such as Instagram number of stories and tagged posts or the time spent using Instagram actively (i.e., chatting, posting) and passively (i.e., viewing, scrolling). Additionally, a replication of the study on data extracted from different social media platforms, such as Twitter or TikTok, may be conducted to verify the generalizability of reported findings. Furthermore, to detect the long-term impact of early life, a longitudinal study to analyze the development of online social skills as an effect of genes and experiences is recommended. Moreover, physiological measures—such as electrocardiography, electromyography, skin conductance, or recordings of neural activity—from peripheral and central nervous systems could be adopted to discover potential differences between online behaviors and offline social responses [93,94]. Moreover, sentiment analysis techniques could be adopted to assess Instagram posts in relation to attachment patterns to predict emotional suffering or mental health conditions [95,96]. Finally, as rs25531 and 5-HTTLPR result in two independent loci for analysis [18], this model could be implemented on 5-HTTLPR.

In conclusion, serotonin transporter gene polymorphisms (rs25531) and early caregiving behaviors play an interactive role in developing online social behavior in Instagram users. Combined with previous evidence, these results offer a contribution to identifying an online social marker in the interplay between genetic and environmental components.

## Figures and Tables

**Figure 1 ijerph-19-05348-f001:**
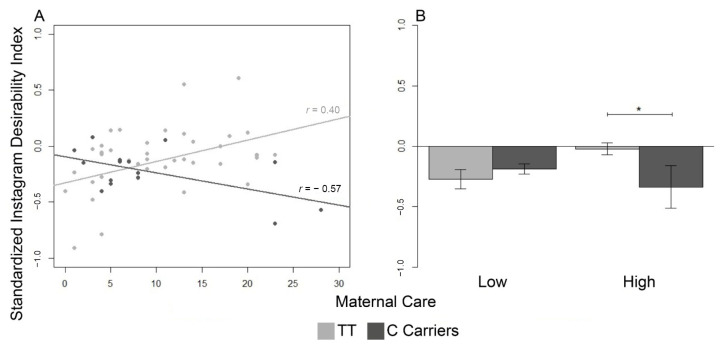
(**A**): Effect of the interaction between genotype and maternal care on the standardized Instagram Social Desirability Index. Correlations between the Social Desirability Index and the reported maternal care. Black circles = C-carriers; gray circles = T/T homozygotes. Lines represent the linear models for C-carriers (black) and T/T homozygotes (grey). Pearson’s *r* values are reported. (**B**): Comparison between Social Desirability Index in C-carriers (black) and T/T homozygotes (gray) divided into high and low maternal care. (* *p* < 0.025).

**Table 1 ijerph-19-05348-t001:** Descriptive statistical summary of the demographic variables: age, sex, and ethnicity (*N* = 57).

Variable	*N*	Percent	Mean	SD
Age	57	-	20.82	1.59
18 y-o	1	1.75	-	-
19 y-o	10	17.54	-	-
20 y-o	20	35.09	-	-
21 y-o	8	14.04	-	-
22 y-o	8	14.04	-	-
23 y-o	6	10.53	-	-
24 y-o	3	5.26	-	-
25 y-o	1	1.75	-	-
**Gender**				
**Level**	* **N** *	**Percent**	**Mean**	**SD**
Males	16	28.07	-	-
Females	41	71.93	-	-
**Ethnicity**				
**Level**	* **N** *	**Percent**	**Mean**	**SD**
Chinese	56	98.25	-	-
Indian	1	1.75	-	-
Chinese Female	40	70.18	-	-
Chinese Male	16	28.07	-	-
Indian Female	1	1.75	-	-
Indian Male	0	0	-	-

**Table 2 ijerph-19-05348-t002:** Descriptive statistical summary of each variable: minimum (Min), first quartile (1st Q), median, mean, third quartile (3rd Q), and maximum (Max).

Variable	Min	1st Q	Median	Mean	3rd Q	Max
**Instagram Variables**						
Posts Number	0.33	0.39	0.49	0.56	0.67	1.19
Followings Number	−1.54	−0.77	−0.12	−0.12	0.52	1.80
Followers Number	−0.80	−0.50	−0.16	−0.11	0.22	1.45
Social Desirability Index	−1.16	−0.28	−0.13	−0.16	−0.03	0.61
**Parental Bonding Dimensions**						
Maternal care	0.00	4.00	8.00	9.97	14.00	28.00
Paternal care	0.00	9.00	14.00	14.49	18.00	30.00
Maternal Overprotection	8.00	18.00	23.00	23.82	30.00	37.00
Paternal Overprotection	11.00	23.00	29.00	27.05	32.00	37.00

**Table 3 ijerph-19-05348-t003:** Mean values in T/T homozygotes and C-carriers divided by PBI dimensions (high or low) on Instagram variables. Standard error means are reported between parentheses.

PBI Dimension	Low/TT	Low/C	High/TT	High/C
**Number of Posts**				
Maternal Care	0.57 (0.05)	0.56 (0.07)	0.55 (0.05)	0.57 (0.12)
Paternal Care	0.56 (0.05)	0.50 (0.04)	0.57 (0.05)	0.64 (0.12)
Maternal Overprotection	0.57 (0.05)	0.63 (0.12)	0.56 (0.04)	0.50 (0.04)
Paternal Overprotection	0.57 (0.05)	0.65 (0.12)	0.55 (0.05)	0.51 (0.06)
**Number of Followings**				
Maternal Care	0.11 (0.19)	−0.27 (0.21)	−0.27 (0.17)	0.01 (0.75)
Paternal Care	−0.19 (0.17)	−0.60 (0.18)	0.01 (0.20)	0.28 (0.41)
Maternal Overprotection	−0.09 (0.19)	−0.034 (0.39)	−0.11 (0.18)	−0.33 (0.30)
Paternal Overprotection	−0.12 (0.16)	0.32 (0.34)	−0.06 (0.22)	−0.53 (0.28)
**Number of Followers**				
Maternal Care	−0.03 (0.13)	−0.19 (0.13)	−0.15 (0.10)	−0.10 (0.38)
Paternal Care	−0.12 (0.11)	−0.29 (0.17)	−0.07 (0.11)	−0.02 (0.20)
Maternal Overprotection	−0.07 (0.12)	−0.15 (0.20)	−0.13 (0.10)	−0.17 (0.19)
Paternal Overprotection	−0.10 (0.11)	0.16 (0.22)	−0.09 (0.12)	−0.38 (0.13)
**Social Desirability Index**				
Maternal Care	−0.27 (0.08)	−0.19 (0.04)	−0.02 (0.05)	−0.34 (0.18)
Paternal Care	−0.21 (0.07)	−0.23 (0.08)	−0.05 (0.06)	−0.23 (0.08)
Maternal Overprotection	−0.11 (0.08)	−0.22 (0.09)	−0.16 (0.06)	−0.23 (0.07)
Paternal Overprotection	−0.14 (0.07)	−0.17 (0.06)	−0.13 (0.06)	−0.26 (0.08)

**Table 4 ijerph-19-05348-t004:** Mean values in male and female participants on Instagram variables. Standard error means are reported between parentheses.

Instagram Variable	Males	Females
Posts Number	0.60 (0.05)	0.55 (0.03)
Followings Number	0.32 (0.23)	−0.30 (0.12)
Followers Number	0.08 (0.12)	−0.19 (0.08)
Social Desirability Index	−0.11 (0.03)	−0.18 (0.05)

## Data Availability

The data of this study can be found in the NTU’s Data repository (DR-NTU Data) at the following address: https://doi.org/10.21979/N9/XIVRWS.

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
