# Peer review of "The Interaction between Serotonin Transporter Allelic Variation and Maternal Care Modulates Instagram Sociability in a Sample of Singaporean Users"

_ijerph, 2022, doi:10.3390/ijerph19095348_

Round 1

Reviewer 1 Report

This paper discuss how the interaction between serotonin transporter allelic variation and maternal care modulates sociability on social networks. The study is interesting and well organized and should be considered for publication. Here are some suggestions for improving the paper.

"Recommender System for Postpartum Depression Monitoring based on Sentiment Analysis." 2020 IEEE International Conference on E-health Networking, Application & Services (HEALTHCOM). IEEE, 2021. DOI: 10.1109/HEALTHCOM49281.2021.9398922

"Postpartum depression prediction through pregnancy data analysis for emotion-aware smart systems." Information Fusion 47 (2019): 23-31. DOI: 10.1016/j.inffus.2018.07.001

1) For this reviewer, some articles of the current state of the art were neglected. A survey can further strengthen the discussion presented in the introduction.

"The interaction between serotonin transporter allelic variation and maternal care modulates sociability on Instagram." (2020). DOI: 10.31234/osf.io/dkvnf

"Adaptive developmental plasticity in rhesus macaques: the serotonin transporter gene interacts with maternal care to affect juvenile social behavior." Proceedings of the Royal Society B 285.1881 (2018): 20180541. DOI: 10.1098/rspb.2018.0541

2) Authors should improve the presentation of the subjects participating in the research. More relevant information is needed.
3) A section on related work should be included. A comparative table is recommended.
4) A high-level discussion on the Social Desirability Index is necessatry.

Reviewer 2 Report

This is a very interesting study on social interactions in social media (Instagram) and their associations with behavioral genetics. However the manuscript still needs some improvements, so I would like to ask the Authors to reply to the following quesions:

- Concerning the participants - could you provide some more precise information, how the students were recruited to the study?

- Concerning the procedure - some reviewers are critical about extracting data from social media for scientific studies. Could you make a comment on that?

- In your recommendations for future studies you say that they should consider a more extensive and multicultural sample to probe potential cross-cultural differences in Instagram behavior. Do you think that other types of social media should be involved as well?

- You say that in future studies physiological measures from peripheral and central nervous systems could be adopted to discover potential differences between online and offline social responses. Could you specify, what kind of measures?

Reviewer 3 Report

Review of submitted manuscript draft:

Title:   The Interaction Between Serotonin Transporter Allelic Variation and Maternal Care Modulates Sociability on Instagram

Reviewer comment

Authors of the manuscript aimed to study a model of “behavioral genetics on Instagram sociability to explore the impact of individual development on the behavior on social networks” as mentioned in abstract. Based of provided hypothesis authors aimed to study social / parenting bonding, relationship and connections to parents or caregivers at early stages of development of the individual and it output to social output and attitudes of the Instagram users.

The manuscript also includes sets of social studies and proband-answering self-questionnaire. Authors used “Social Desirability Index” and “The Parental Bonding Instrument (PBI)”; found “gene-environment interaction” of the users of Instagram, as they hypothesized. At the presented manuscript authors used gene expression studies (PCR) and DNA sequencing technique focused on serotonin transporter gene (SLC6A4), at least it looks like. In the methods sections authors provided just primer sequence “(forward 5’-GGCGTTGCCGCTCTGAATGC-3’ and reverse 149 5’-GAGGGACTGAGCTGGACAACCAC-3’)”. Genetic analysis has been performed via external company using commercial sample kits.

Research approach is not clear at the moment. Primary purpose of this study is understandable, in the time of prevalent usage of the social networks with all accessible pros and cons, but unfortunately the final message persisting unclear.

Authors of the manuscript are focusing their research on the social behavior and social interaction, but there is any section related to, for example Oxytocin system, hypothalamus, hippocampus, or other areas responsible for social bonding and social interactions.

The conclusion section is presented, and as the authors mentioned, this manuscript has some serious limitations. Some of them are minor, but unfortunately some must be seriously improved, or some ideas would have to be reengineered. The Main goal of the study is not effectively described with possible thought if full potential of intended research has been used. Abbreviation description – as separate section of the main text – would be appreciated.

Although potentially interesting and beneficial study on the relation between polymorphism of one of the major neurotransmitters related to social behavior and early stage of individuals brain development, together with parental bonding - resulted into late social behavior and interaction of the Instagram users – hasn’t reach its full potential.

Reviewer recommendations:

  1. Abstract of the submitted manuscript is mentioning 57 participant (Instagram users) and main body of manuscript but Section 2.1 mention “Sixty-one (N = 61) young adults”. Could you please address this disbalance? Why 4 participants have been excluded? What “technical issues” means in the view of presented study?
  2. As the same part mentioning “age lower than 30 years old, and (iii) owning and using an Instagram account.” What does exactly mean? Please provide proper data about age distribution.
  3. It would be also appreciated if authors will provide more information related to the participants, they socio-geographical status, and more. Also, information about usage of the social app of interest would be appreciated. How extensive and for how long are participants actively employed by using social app?
  4. Did authors supervise Parental Bonding Instrument questionnaire at any way? Are those collected data valuable? Was any “Lie Score Precaution” employed?
  5. Authors are using term “Instagram people” without any explanation.
  6. Authors of the manuscript assume that users of the social apps – Instagram in this case – are behaving as an uniform mass of consumers of the social content. All the focus of the study is based on Follower-Following; and Post-Repost-Comment principle. Authors are also assuming that some followed content will automatically become a role model of the behavior. How did you mean this? If hypothetically there only one strict content consumer focused on the plats or animal, hoe will you elaborate this outlier?
  7. Please elaborate limitation of this study relate to using 57 (61) participants. Are you able to generalize your findings and conclusions globally?
  8. In the section 3 of the main text of the manuscript authors are referring: “Instagram variables: Preliminary Results”. Please make clear if this study is research article or preliminary short report.
  9. At the whole text, authors are focusing on the social behavior and social interaction, but there is no word related to, for example Oxytocin system, hypothalamus, hippocampus, or other areas responsible for social bonding and social interactions. Are you able to provide this section, or at least discuss your finding in the Conclusion and discussion section of the manuscript?

Round 2

Reviewer 1 Report

The authors considered all my suggestions.  

Author Response

We want to thank the reviewer for the suggestions.

Reviewer 3 Report

To Whom It May Concern, Authors of the manuscript,  

Thank you for your responses for my comments and recommendations.

Your answers have been adequate, but unfortunately not fully sufficient. I would like to ask few more question or address some recommendation:

  1. As you declare, participant’s socio-demographical status has been described as “Number of Chinese and Indian participants”. From presented data, the actual ratio has been 56 Chinese to 1 Indian participant. This can’t and shouldn’t be presented as normal distribution of general pool of Instagram users. Or is it representing a general groups distribution at the university area, campus, pool of students related to targeted age? If yes, please provide more data for evaluation.
  2. Presented ratio of 56 Chinese to 1 Indian participant. need to me mentioned as serious limitation of presented study. I would highly recommend considering change of the title of the presented study itself in this case.
  3. Did all the participant of the presented study used the same version of targeted Instagram application, and the cell phone software? Did you collect these data as well? Will it be worth, based on your findings, consider this aspect as well?
  4. Is there any publicly know differences or another geographically optimized version of Instagram application used by participants in comparison with general international Instagram application? Were there any possible limitations regarding to general use and content access? If yes, or no, would it be considered as a limitation of study – eventually will it stand for any consideration as a limitation?
  5. Based on presented data, participants were recruited from pool of students at Nanyang Technological University of Singapore and were awarded with one credit as contribution of the Psychology Assessment module. Can you provide valid data of form of recruitment, inclusion / exclusion criteria form general students’ population? The gain profit equal to one credit – what was the potential value if this award? Can it be considered as a major motivation to participate as “volunteer”?
  6. Based on provided data – participants from 18 to 25 years of age have been use. Many of known changes and maturation process can occur in the neuro-endocrine and psychological-social system at targeted age on the border line between puberty-adolescent-adulthood. Were any of these aspects considered?
  7. Unfortunately, authors declaration: “As regards age lower than 30 y-o, the decision was made in order to have a homogeneous group of participants having about the same age to allow comparison.” Can’t be consider as valuable or even correct as a research approach (age 18-25 as participant pool). Please reconsider this standing.
  8. We all can agree with statement “Following, instead, is a type of passive communication, for the user is exposed to the contents selected by others…*” From the other hand, based of very unclear, vague and many times even unknown content offer regulated by social network algorithm an AI we can’t agree with statement “*…but that the users actively decide to watch.” at any cost. This statement could be true or value only if authors are able to analyze and provide data that participants have only and strictly interact with real followed social network member (alive, real, friend - which is followed by participant willingly) and no others reel, advertisement, paid promotions and other “recommended” content selected by social network itself. Please make clear this area and or make sure it will be mentioned as a limitation of the presented study.
  9. There is no further comment related to genetic analysis, methodology, scientifical approach and results presentation on this part of the presented manuscript.

Round 3

Reviewer 3 Report

I would like to thank all authors for their cooperation.